# Structural Analysis and Dynamic Processes of the Transmembrane Segment Inside Different Micellar Environments—Implications for the TM4 Fragment of the Bilitranslocase Protein

**DOI:** 10.3390/ijms20174172

**Published:** 2019-08-26

**Authors:** Kosma Szutkowski, Emilia Sikorska, Iulia Bakanovych, Amrita Roy Choudhury, Andrej Perdih, Stefan Jurga, Marjana Novič, Igor Zhukov

**Affiliations:** 1NanoBioMedical Centre, Adam Mickiewicz University, Wszechnicy Piastowskiej 3, 61-614 Poznań, Poland; 2Faculty of Chemistry, University of Gdańsk, Wita Stwosza 63, 80-308 Gdańsk, Poland; 3Institute of High Technologies, Taras Shevchenko National University, Volodymyrska 54, 01-601 Kyiv, Ukraine; 4National Institute of Chemistry, Hajdrihova 19, 1001 Ljubljana, Slovenia; 5Institute of Biochemistry and Biophysics, Polish Academy of Sciences, Pawińskiego 5a, 02-106 Warsaw, Poland

**Keywords:** bilitranslocase, transmembrane peptide, NMR spectroscopy, ^31^P CPMG

## Abstract

The transmembrane (TM) proteins are gateways for molecular transport across the cell membrane that are often selected as potential targets for drug design. The bilitranslocase (BTL) protein facilitates the uptake of various anions, such as bilirubin, from the blood into the liver cells. As previously established, there are four hydrophobic transmembrane segments (TM1–TM4), which constitute the structure of the transmembrane channel of the BTL protein. In our previous studies, the 3D high-resolution structure of the TM2 and TM3 transmembrane fragments of the BTL in sodium dodecyl sulfate (SDS) micellar media were solved using Nuclear Magnetic Resonance (NMR) spectroscopy and molecular dynamics simulations (MD). The high-resolution 3D structure of the fourth transmembrane region (TM4) of the BTL was evaluated using NMR spectroscopy in two different micellar media, anionic SDS and zwitterionic DPC (dodecylphosphocholine). The presented experimental data revealed the existence of an α-helical conformation in the central part of the TM4 in both micellar media. In the case of SDS surfactant, the α-helical conformation is observed for the Pro258–Asn269 region. The use of the zwitterionic DPC micelle leads to the formation of an amphipathic α-helix, which is characterized by the extension of the central α-helix in the TM4 fragment to Phe257–Thr271. The complex character of the dynamic processes in the TM4 peptide within both surfactants was analyzed based on the relaxation data acquired on ${}^{15}$N and ${}^{31}$P isotopes. Contrary to previously published and present observations in the SDS micelle, the zwitterionic DPC environment leads to intensive low-frequency molecular dynamic processes in the TM4 fragment.

## 1. Introduction

Membrane proteins that regulate cell-membrane trafficking are being extensively studied as potential drug targets due to their ability to act as tumor biomarkers as well as effective cancer drug transporters [1,2]. Consequently, there is a strong interest in the family of organic anion transporter proteins (OATPs), which are often altered in malignant tissues. Screening tumors for OATP expression may enable an OATP-targeted therapy with higher efficacy and most importantly, decrease side effects relative to the current anti-cancer therapies. In our previous studies we determined the transmembrane protein bilitranslocase [3] (BTL) (UniProt O88750, TCDB 2.A.65) to be a potential drug target as it exhibits partial functional similarity to OATPs [4]. The BTL protein regulates the transport of organic anions such as bilirubin through the liver cell membrane [5]. Thus, it plays an important role in both human pathology [6] and drug delivery [7]. Although the primary structure and biological functions of the BTL have been established previously [3,4,5,6,7,8,9,10,11,12,13], the secondary and tertiary structures of the BTL are still not known. In particular, the BTL has no sequence homologs in UniProt database. Thus, conclusion about its transport mechanism and as well as its structural features cannot be easily extrapolated from the available data originating from other transport proteins. Based on the homology-independent considerations, it has been predicted that BTL is comprised of four TM α-helices [14]. So far, the existence of the two transmembrane α-helical structures in SDS media for the TM2 and TM3 fragments has already been validated with NMR spectroscopy [15,16]. Now we extend our structural and dynamical studies to the fourth transmembrane region of bilitranslocase. Our previous studies also detected the existence of slow molecular motions specific for the TM2:TM3 pairs due to *cis-trans* isomerization of peptide bonds including the central prolines (Pro85 in TM2 and Pro231 in TM3) [17].

We surmised that the fourth segment TM4 is substantially more hydrophobic compared to the TM2 and TM3 segments. This makes the TM4 particularly interesting for exploring of the residue-specific interactions within the different micellar media. Membrane proteins still remain a major challenge for structural biology. Several techniques are used in structural studies of hydrophobic moieties in the lipid media [18]. The majority of 3D structures available in the PDB databank have been determined by X-ray crystallography. Nevertheless, NMR spectroscopy yields critical experimental data to explore conformational molecular dynamics of flexible segments [19] or *cis–trans* heterogeneity [20], which after supplemented with computational analysis can expand our knowledge of the transmembrane protein structures [21]. It includes sequence–dependent predictions of transmembrane regions, their stability and interactions, as well as molecular dynamics simulations, coarse-grain simulations and other stochastic methods.

The TM4 hydrophobic segment of the BTL protein has been identified using a software-based predictor developed earlier [14,22]. The predictions were supported with molecular dynamics simulations in dipalmitoylphosphatidylcholine (DPPC) lipid. Performing structural analysis of the two possible lengths of the TM4 segment—defined as TM4—predicted by our initial algorithm and TM4A obtained by its statistically improved version in aqueous solutions of DPPC is complicated due to the inherent instability of phospholipid/water systems where the size distribution, as well as the path-dependent morphology, may give ambiguous results [23].

The micelar environments are formed by the components with different affinity to a solvent— hydrophilic ‘head’ and hydrophobic ‘tail’. Combined in one particle they allow forming an aggregate in aqueous solution, which can be considered as a simple amphiphilic environment. The negatively charged anionic SDS micelle is used as a mimetic of prokaryotic cell membrane which is still used for structural studies of transmembrane peptides. The zwitterionic dodecylphosphocholine (DPC), belonged to the class of alkyl phosphocholine detergents constitutes a better model for the eukaryotic cell membrane [24] compared to SDS.

Helical membrane proteins (like BTL) demonstrate lower stability in zwitterionic lipid media due to the highly hydrophobic composition of transmembrane fragments [25]. As recently established, the residues with lack (Gly) or short side chains (Pro, Ala) are playing a key role in stabilization of the 3D structure and define the proper orientation of TM helices [26]. Such residues are usually highly conserved in the specific positions and constitute potential binding sites for other TM helices, which are extremely important for the creation of weak electrostatic interactions and hydrogen bonds necessary for the stabilization of the whole helix bundle [25,26].

Here we presented the results of the structural analysis of the TM4 fragment in two surfactants using multi-dimensional NMR techniques. First, we performed a structural analysis of the TM4 segment in SDS micelle, similarly to our previous studies [15,16,17]. Also, the zwitterionic DPC micelar media were used to extend our knowledge about 3D structure fourths TM fragment of the BTL protein in the eukaryotic cell membrane. In our studies, NMR experiments were performed in perdeuterated variants of the SDS-$d_{25}$ and DPC-$d_{38}$ micelle, which facilitates the direct observation of the self-diffusion on the ${}^{2}$H and ${}^{31}$P isotopes and allows a straightforward comparison with our previous results obtained for the TM2 and TM3 peptides. To explore molecular dynamics processes, we introduced a ${}^{15}$N-labeled Ala261 into the central part of the TM4 segment, to acquire the ${}^{15}$N relaxation data ($R_{1}$ and $R_{2}$ relaxation rates). The analysis of the internal molecular dynamics of the TM4 segment was carried out with ${}^{15}$N and ${}^{31}$P relaxometry. To the best of our knowledge, the different character of the α-helical peptide segments inside the DPC interior together with unexpected dynamic processes, have not yet been reported in the literature and may be relevant for the investigation of membrane proteins in a surfactant environment.

## 2. Results

### 2.1. Prediction of the TM4 Transmembrane Region

The transmembrane region of BTL protein was predicted using several algorithms (Appendix A). The first algorithm, PredαTM, predicts the initial transmembrane regions based on the primary sequence of a protein [14,22]. The initial prediction is further refined using the position-specific amino acid statistical information regarding the presence of the residues in α-helical transmembrane regions [14,22]. The final transmembrane regions of BTL with statistically favored terminals was predicted at residues 24–48 (TM1A), 75–94 (TM2A), 220–238 (TM3A) and 254–276 (TM4A). It must be noted that the statistical scoring method was introduced only to fine-tune the predicted transmembrane regions—the core residues of the transmembrane regions remained essentially similar. The incorporation of the amino acid preference patterns influences the terminal residues only in such a way that they are more statistically favored. According to our findings, the PredαTM algorithm first determined the existence of four stretches out of ten or more consecutive segments, as a transmembrane region. These segments span over residues 16–53 (19 segments), 65–103 (20 segments), 213–246 (15 segments) and 250–285 (17 segments) [22]. The central residues of these predicted transmembrane stretches, 24–45 (TM1), 73–95 (TM2), 221–238 (TM3) and 258–277 (TM4) were considered for final transmembrane region prediction [22]. Along with PredαTM, several other state-of-art alpha transmembrane region prediction algorithms were used to predict the transmembrane regions of BTL and a consensus prediction was thus generated. Only two algorithms predicted four transmembrane regions for BTL, namely TMpred [27] and TopPred II [28]. The algorithms PRED-TMR [29], MemBrain [30] and Philius [31] predicted three transmembrane regions: TM1, TM2 and TM4 while HMMTOP [32], SCAMPI [33] and TOPCONS [34] predicted other combinations of transmembrane regions: TM1, TM3 and TM4. The rest of the algorithms considered here predicted only two transmembrane regions TM1 and TM4. Nonetheless, SOUSI [35] classifies BTL as a globular protein. Our finding is that the core residues of all the predicted transmembrane regions are similar to the transmembrane regions predicted using PredαTM.

In conclusion, while most predictors have failed to identify the TM2 and TM3 transmembrane regions, the TM1 and TM4 were predicted most consistently. Interestingly, the content of leucine in the TM1 and TM4 segments was 28% and 26% respectively. In both cases, the predicted content of Leu is significantly higher than the average for α transmembrane regions. On the other hand, the TM2 and the TM3 transmembrane regions show a significantly lower occurrence of Leu, 15% and 10.5%, respectively. Accordingly, we have determined two most probable TM4 configurations depicted as TM4 and TM4A. MD simulations further confirmed the stability of predicted segments in the DPPC membrane.

### 2.2. Assessment of the TM4/TM4A Stability in DPPC From MD Simulations

The initial 3D configurations of the simulations of TM4 and TM4A in DPPC are shown in Figure 1. MD trajectories of 20 ns duration were analyzed for overall conformational change. The secondary structures were visualized using the STRIDE algorithm within VMD [36]. We provide the animations for both TM4 and TM4A systems in the Supporting Information. We have performed MD simulations for the initially predicted BTL TM4 transmembrane regions spanning over residues 258–277 and residues 254–276 for the TM4A, respectively [15,16]. Furthermore, we have added two additional amino acids on the C-terminal and N-terminal ends of each of the α-helices to explore an extended conformational space and soften the terminal boundary at the same time. The water molecules mimicked the aqueous environment of the extra- and intracellular compartments.

Initial visual inspection has shown that both sequences TM4 and TM4A preserved α-helical conformations in the secondary structure Appendix A. To further quantify this qualitative observation RMSD values for all backbone atoms generated in the α-helical positions were calculated. For the RMSD comparison backbone atoms of those residues that were predicted to form the α-helix by the chemometric predictions were used [15,16]. The obtained RMSD parameters fully corroborated the α-helix conformations of both TM4 and TM4A and did not undergo substantial changes during the MD simulations (Appendix A). The average RMSD value for the α-helix of TM4 was 0.65 ± 0.10 Å and of TM4A 0.65 ± 0.20 Å thus showing a high level of structural integrity during the simulation. In comparison to the TM2 and TM3 helices, the presence of the Pro258 residue in the predicted transmembrane sequences did not induce a kink in the α-helical conformation which can be attributed to the Pro258 at the beginning (TM4) or amino acid residues away from the predicted helix beginning (TM4A). We believe that in this position, Pro258 cannot significantly interfere with the overall structure as was observed previously for the TM2 and TM3 structures where corresponding proline residues (Pro85 in TM2 and Pro231 in TM3) were positioned in the middle of the predicted helices [15,16].

Furthermore, we have determined the distribution of the torsion angles ϕ and ψ for both systems. As previously, we used three dimensional Ramachandran plots shown in Appendix A, which is useful for viewing the frequency distribution of all dihedral backbone angle values observed during the MD simulation. Only specific values for torsion angles are allowed in the α-helical conformation and this provided another way of assessing the integrity of the α-helical conformation. It is known that α-helices in TM proteins typically adopt backbone ϕ and ψ dihedral angles around −60° and −45°, respectively [37]. In our study, the ϕ and ψ angles of those residues that were predicted to be in α-helix showed a very uniform and stable distribution around the values expected for the α-helix conformation. The VMD program was further used to produce two-dimensional plots. We plotted the predicted secondary structures against simulation time for each simulation frame. The overall plots for both helices in Appendix A. The α-helix structure again appears to be fully stable in both systems.

The overall findings of our initial MD assessment demonstrated that BTL sequences TM4 and TM4A predicted by the chemometric approach could adopt a stable α-helical conformation when inserted into the DPPC membrane during the 20 ns MD simulation. A small difference between both analyzed sequences was in their length, the 258–277 sequence in TM4 was slightly shorter than the 254–276 sequence in TM4A. Simulation results indicated that a more extended TM4 BTL sequence which encompasses the whole shorter TM4 except for residue 277, is sufficiently stable in the α-helical conformation. Although these are reasonably long MD simulation runs, such a simulation time still does not enable full coverage of the conformational space for an unambiguous quantitative stability assessment of the investigated transmembrane helix [16]. In order to to provide experimental evidence for the initial chemometric/simulation experiments KKK${}^{254}$PNIFPLIACILLLSMNSTLSFS${}^{276}$ TM4A BTL sequence was selected for further and more detailed structural investigations.

### 2.3. CD Spectroscopy Confirm Existence α-Helical Conformation for the TM4 Fragment in SDS and DPC Surfactants

The existence of stable α-helix as a secondary structure motif for the TM4 peptide in both—SDS and DPC micelles—was confirmed experimentally by collection CD spectra in various media (Appendix A). In aqueous solution (without surfactant) CD data reveals a minimum at ca. 200 nm, which is typical for unordered structures. As expected, the highest α-helical content (∼60%) is observed in the TFE solution, which is a well-known ‘helix–inducing’ solvent [38]. Positioning of the TM4 fragment in DPC or SDS micelles resulted to folding peptide in a α-helical conformation with two characteristic minima observed at 208 and 222 nm [39,40]. Quantitative analysis of the CD spectra shown the fraction of α-helical content reaches 46% and 42% in DPC and SDS micelles, respectively. We expect the larger errors for the estimation of α-helical content due to a small number of reference structures (only 13 membrane proteins available in the SMP50 database) [41]. Moreover, the effect of ‘absorption flattening’ appeared due to non-uniform distribution of chromophores within the sample can additionally disturbed the calculations [42].

### 2.4. Solution High-Resolution 3D Structure of the TM4 Fragment in SDS-$d_{25}$ and DPC-$d_{38}$ Micelle by Means NMR Spectroscopy

Assignments of ${}^{1}$H, ${}^{15}$N and ${}^{13}$C resonances for TM4 segment in SDS-*d*${}_{25}$ and DPC-*d*${}_{38}$ micelles were carried out based on homonuclear 2D TOCSY and 2D NOESY NMR experiments [43] supplemented with heteronuclear ${}^{1}$H–${}^{13}$C and ${}^{1}$H–${}^{15}$N HSQC experiments acquired on the natural abundance of ${}^{13}$C and ${}^{15}$N isotopes. An initial inspection of experimental data demonstrated the existence of ${}^{1}$H${}^{N}$–${}^{1}$H${}_{\left(i,i+3\right)}^{N}$ contacts together with ${}^{1}$H${}^{N}$–${}^{1}$H${}_{\left(i,i+3\right)}^{\alpha }$ cross-peaks characteristic for α-helical conformation (Appendix A).

The high-resolution 3D structures of the TM4 peptide were evaluated with CYANA software [44] based on 258 (124 intra-residual, 113 sequential and 21 medium range) and 278 (170 intra-residual, 88 sequential and 20 medium range) distance constraints yielded from 2D ${}^{1}$H–${}^{1}$H NOESY spectra recorded in the SDS-*d*${}_{25}$ and DPC-*d*${}_{38}$ micelle, respectively. These data were supported by the restraints for the backbone ϕ and ψ torsion angles extracted with the TALOSn software [45] using assigned ${}^{1}$H, ${}^{13}$C and ${}^{15}$N chemical shifts. Finally, 12 distance constraints for six hydrogen bonds (for SDS-$d_{25}$) or 16 distance constraints for eight hydrogen bonds (for DPC-$d_{38}$) were defined based on the geometric criteria and applied only in the refinement stage of the structure calculations. The experimental data together with the analysis of the quality of the produced ensemble of 20 TM4 structures in both surfactants are summarized in Appendix A.

The 3D structure of the TM4 peptide evaluated in SDS-$d_{25}$ and DPC-$d_{38}$ micelle presented as an α-helical conformation for the central part of the TM4 peptide and less defined structure on both the N- and C-termini (Figure 2A,B), in agreement with our predictions and with computer simulations. The obtained results demonstrate that the 3D structure of the TM4 is folded in a stable α-helix, similarly to the 3D structures that have been previously observed for the TM2 and TM3 segments in SDS-$d_{25}$ surfactant [15,16]. Superposition of the TM4 structures in SDS and DPC micelle yielded r.m.s.d. of 0.79 Å over residues located in the α-helix (Appendix A).

Despite the structural similarity of the TM4 fragment in the SDS and DPC micelle, some structural alterations can be noted. In particular, replacement of the anionic SDS with the zwitterionic DPC surfactant resulted in the expansion of the α-helix by one turn, from Pro258–Asn269 up to Phe267–Thr271 (Figure 2B). The values of the ϕ and ψ backbone torsion angles for residues Phe271 (N-termini) and Ser270–Leu272 (C-termini) clearly show α-helical conformation in the DPC-$d_{38}$ micelle (Appendix A).

The chemical shifts perturbations (CSP) between SDS and DPC media show regular upfield and downfield shifts for ${}^{1}$H${}^{N}$ resonances (Figure 3A). Taking into account position in the micelle, it reveals an upfield shift for the residues localized on one side of the α-helix, suggesting a more hydrophobic environment compared to the corresponded position in the SDS-$d_{25}$ (Figure 3B).

### 2.5. Translational Mobility of TM4 Fragments and Micellar Media with ${}^{1}$H, ${}^{2}$H and ${}^{31}$P PGSE NMR

The impact of anionic or zwitterionic surfactant on translational mobility of the TM4 fragment was studied using diffusion NMR experiments. The values of the translational diffusion coefficient ($D_{tr}$) were extracted from the ${}^{1}$H experimental data for the TM4 peptide in SDS-$d_{25}$ and DPC-$d_{38}$ micellar environments (Appendix A). Assuming that contained TM4 peptide SDS and DPC micelles are presented as spheres or ellipsoids with a small fraction of an anisotropy, the hydrodynamic radii ($R_{h}$) can be estimated based on the Stokes-Einstein equation [46]:$$R_{h}=\frac{k_{B}T}{6\pi \eta D_{tr}}$$
where $k_{B}$ is a Boltzmann constant, *T* is the absolute temperature, η is the viscosity of a solvent. The calculated $R_{h}$ are within the range of 33–34 Å (Appendix A), which is in good agreement with our previous results obtained for TM2 and TM3 segments in SDS-$d_{25}$ micelle at 303 K [17].

The diffusion analysis of TM4 in different surfactants could be expanded by NMR experiments on other isotopes. In particular, the application of deuterated forms of SDS-$d_{25}$ and DPC-$d_{38}$ surfactants allows characterizing translational motions of the micelle by measuring the $D_{tr}$ values on ${}^{2}$H or ${}^{31}$P (specifically for DPC-$d_{38}$) isotopes. Deuterium measurements exhibited a slight increase in the $R_{h}$ values (up to 37–38 Å) for both surfactants (Figure 4).

The $D_{tr}$ values obtained for ${}^{31}$P isotope reveals an increased $R_{h}$ value for DPC-$d_{38}$ even more, up to 40 Å (Appendix A). Taking into account localization of the phosphate group close to the surface of the DPC-$d_{38}$ micelle, the observed effect could be explained by the deviation from the spherical shape of micelles [47], which becomes more pronounced for the nucleus at the longer distance away from the center of mass.

### 2.6. Probing the Molecular Dynamics of TM4 Segment Inside SDS-$d_{25}$ and DPC-$d_{38}$ Micelles by ${}^{15}$N NMR Relaxation

The relaxation data measured for the ${}^{15}$N nuclei in the amide group provide useful information about the backbone dynamics in peptides and proteins. A ${}^{15}$N-labeled Ala261 in the TM4 peptide enabled exploration of the molecular dynamic processes in the α-helical region (Figure 5). Compared to previously reported data for the TM2 and TM3 fragments from the BTL protein (1.52/1.33 s${}^{-1}$ for Ala80 and Ala88 in TM2 and 1.49/1.42 (s${}^{-1}$) for Ala225 and Ala233 in TM3 [17]), the extracted $R_{1}$ values for the TM4 segment show decrease $R_{1}$ relaxation rate. The substantially faster spin-spin relaxation reflected in the increased $R_{2}$ relaxation rate facilitates a different kind of dynamic process in the TM4 fragment compared to the previously studied TM2 and TM3. The calculated overall correlation time ($\tau_{c}$) from the $R_{2}/R_{1}$ ratio under the assumption of isotropic rotation yields a value of 7.3 ns, which is substantially longer than the 4.9 ns reported for TM2 and TM3 peptides [17]. As we noted before, the TM4 fragment is much more hydrophobic compared to the previously studied TM2 and TM3 fragments which is reflected in stronger hydrophobic interactions with the SDS micelle. An analysis performed using Yasara software showed grow up the number of contacts and energy of hydrophobic interactions increased from 6.6 kJ/mol (for TM2) and 8.4 kJ/mol (for TM3) up to 16.6 kJ/mol for TM4 (Appendix A).

When we compare the $R_{1}$ and $R_{2}$ relaxation rates obtained for the TM4 segment in SDS-$d_{25}$ and DPC-$d_{38}$ micelles, the obtained experimental data revealed a two-fold lower longitudinal relaxation rate $R_{1}$ and nearly three times higher transverse relaxation rate $R_{2}$ concerning the corresponding values obtained in the SDS-$d_{25}$ micelle (Table 1). The results presented the different layout of dynamic processes for the TM4 backbone. The $R_{2}$ relaxation rate is mostly responsible for the slow dynamic processes, observed within a 10${}^{-3}$–10${}^{-6}$ (s) time frame.

The measured $R_{1}$ and $R_{2}$ relaxation rates evaluated for ${}^{15}$N in Ala261 allow us to estimate the rotational correlation time ($\tau_{c}$). For TM4 in DPC-$d_{38}$, the $\tau_{c}$ was around 21 ns which is three times longer than the corresponding value obtained for TM4 segment in SDS-$d_{25}$ surfactant (Table 1). Obtained results might be consistent with $\tau_{c}$ values obtained for proteins of molecular mass 40–50 kDa [48] but TM4 fragment is much smaller. At the same time, we observe no differences in the diffusion experiments (Appendix A). Therefore, we conclude that R${}_{2}$ values are altered due to the interaction of the TM4 segment with DPC micelle. In the next section, we demonstrate the application of ${}^{31}$P NMR to probe the slow dynamic processes of the TM4 fragment inside DPC-$d_{38}$ micelle.

### 2.7. Snorkelling Interactions between N-Terminal Lysines and Phosphate Groups Observed in the DPC-$d_{38}$ Micelle by ${}^{31}$P NMR Relaxation Experiments

The ${}^{31}$P isotope, whose relaxation is controlled by the CSA (Chemical Shift Anisotropy) mechanism [49] is an excellent probe for studying dynamic processes in the hydrophilic part of the DPC surfactant. Lysine residue contains a protonated -NH${}_{3}^{+}$ group at the end of their side chain, which can interact with negatively charged phosphate groups -PO${}_{4}^{-}$ presenting in the DPC surfactant. The results of such interactions are snorkeling of the amine groups at the micelle surface (Appendix A) [50], which is detected in dynamic studies by ${}^{31}$P $R_{1}$ and $R_{2}$ relaxation rates (Figure 6A,B). The acquired data reveals a stable $R_{1}$ relaxation rate upon adsorption of TM4 segment into the DPC micelle; however, the transverse relaxation rate $R_{2}$ changed dramatically (Figure 6B and Appendix A).

To further explore low-frequency processes in the DPC-$d_{38}$ micelle, we performed a ${}^{31}$P CPMG experiment as $R_{2}\left(\tau_{cp}\right)$ dispersion versus pulse frequency [51]. The presented technique is susceptible to slow exchangeable processes where spins undergo translations between two or more unique sites characterized by a substantial difference in the amplitude and nonhomogeneity of the local magnetic fields, different residence times and populations. The relaxation rates $R_{2}\left(\tau_{cp}\right)$, where $\tau_{cp}$ is the π–π spacing in CPMG experiment, were analyzed using the Carver-Richards (CR) model [51,52].

A CPMG dispersion experiment recorded for an ‘empty’ (without TM4) DPC-$d_{38}$ micelle demonstrated the absence of exchange phenomena (Figure 6C). Inserting the TM4 fragment resulted in a strong conformational exchange dynamic existing for the phosphate group of the DPC-$d_{38}$ micelle (Figure 6C). An analysis of the experimental data was performed assuming two-site exchange motion where the frequency difference Δω in the is the ${}^{31}$P chemical shift difference between the two states of the DPC-$d_{38}$ molecules inside a micelle and can be applied to conformational exchange processes [53]. The applied procedure showed the existence of two conformational states with spin populations of $P_{a}=0.972$ and $P_{b}=0.028$ (See Appendix A for details). The relaxation rates calculated for both conformations are $R_{2a}$ = 31.25 s${}^{-1}$ and $R_{2b}$ = 5.26 s${}^{-1}$. The frequency differences $\Delta \omega =5.2\times 10^{3}$ and the corresponding pseudo-first order exchange rate is $k_{ex}=253$ s${}^{-1}$ (Figure 6C). We believe that the second state populated as 2.8 % is described phosphate groups directly interacting with the lysines side chains in a snorkeling regime in the DPC-$d_{38}$ micelle. Nevertheless, this group of ${}^{31}$P nuclei is enough to create an effect that is visible by the CPMG experiment.

Although we can provide other possible origins of the dispersion curve, such exchange of the TM4 peptide between different micelles, given the circumstances, we believe that such a process will be slow in the NMR time-scale so that two-exponential decays have to be observable in ${}^{15}$N relaxation data (Figure 5B). This finding is not the case since the decays are single-exponential.

## 3. Discussion

### 3.1. The Change from an Anionic to a Zwitterionic Environment Results in a Different Character of the α-Helix and Backbone Dynamics

The central part of the TM4 transmembrane region contains only hydrophobic residues: Pro254, Phe257, Pro258, Leu259, Ile260, Ile263, Leu264, Leu265, Leu266, Leu272 and Phe275 that formed a large hydrophobic surface in the central part of the α-helical structure (Figure 7). The structural analysis reveals the existence of strong hydrophobic interactions between that part of the TM4 fragment and SDS and DPC micellar environments especially in the fragment ${}^{263}$Ile-Leu-Leu-Leu${}^{266}$ (Appendix A). However, a close view of the solved 3D structures demonstrated a substantial difference in packing of the TM4 fragment in the SDS and DPC surfactant. For the TM4 fragment in SDS-$d_{25}$ micelle, the calculated hydrophobic surface of the central part is equal to 1148 Å${}^{2}$, which is located around the entire α-helix (Figure 7A). The zwitterionic environment for the TM4 peptide enables a formation of a channel inside the DPC-$d_{38}$ micelle (Figure 7B) resulting in a slight decrease of the hydrophobic surface to 1015 Å${}^{2}$. The calculated energy of hydrophobic interactions is equal to 16.6 (kJ/mol) and 14.3 (kJ/mol) for the TM4 segment in SDS and DPC lipid media, respectively (Appendix A).

At the same time, the content of α-helical conformation is increased as can be observed from CD measurements (Appendix A), evaluated 3D structures by NMR spectroscopy and molecular dynamic simulations performed by the Yasara software (Appendix A).

The hydrophobic contacts for residues side chains in the TM4 fragment were evaluated with the Radial Distribution Function (RDF) calculated using the Ptraj program [54]. An analysis of the RDF data showed regular changes in the hydrophobic environment between both types of the micelle (Appendix A). The difference in hydrophobic surface facilitates an opponent character of the α-helix in the case of the DPC-$d_{38}$ compared to the SDS-$d_{25}$ micelle, which are visualized by the wheel plot (Figure 7C,D). The TM4 segment inside the SDS-$d_{25}$ micelle characterized the positioning of the hydrophobic residues around the whole α-helix, forming strong hydrophobic contacts with the surfactant. On the other hand, the TM4 fragment inside the DPC-$d_{38}$ resulted in a formation of the clearly visible hydrophobic surface only on one side of α-helix and hydrophilic part on the other side. Our results suggest the different character of the central α-helix in the TM4 fragment changes from hydrophobic in anionic SDS to amphipathic in the zwitterionic DPC micelar environment.

### 3.2. A Comparison with Previously Evaluated Structures of the TM2 and TM3 Transmembrane Fragments in SDS Micelle

A comparison of the solved 3D structure with previously evaluated TM2 and TM3 transmembrane regions show some specific features that can suggest of the role of the TM segments in the spatial organization of the BTL channel. The TM2 fragment is presented as a helix–loop–helix motif (Cys76–Leu82, Phe86–Leu99) with the Gln83–Pro85 loop accessible to solvent (Appendix A) [16]. Analysis of the TM3 region reveals the α-helical structure for the C-terminal part (Ala225–Thr237) of the peptide (Appendix A) [15]. The ${}^{15}$N relaxation data acquired for the ${}^{15}$N-labeled alanines demonstrated in the TM3 segment existence of low-frequency dynamic processes related to the *cis/trans* isomerization around the Leu230–Pro231 peptide bond. The positioning of Leu230–Pro231 in TM3 is correlated with the position of Ser84–Pro85 in the TM2 region so the *cis/trans* isomerization dynamic process could facilitate the process of the conversion between ‘open’ and ‘closed’ states of the BTL channel [17]. The TM4 segment analyzed in this study exhibited a long α-helical structure included a central highly hydrophobic part of the peptide (Appendix A).

There are four TM regions defined for the entire BTL protein which have α-helical conformation. The mutual orientation inside the helix bundle is defined by weak electrostatic interactions and hydrogen bonds appeared when the residues with short side chains (prolines, alanines) and especially glycines expose polypeptide backbone [25]. The sequence of the previously studied TM2 fragment (Ser73–Leu99) contains two glycines (Gly89 and Gly92), one proline (Pro85) and two alanines (Ala80, Ala88). The TM3 segment contain one glycine (Gly226), one proline (Pro231) and two alanines (Ala225 and Ala233). Finally, the TM4 segment studied in this work has only one alanine (Ala261) and one proline (Pro258), which suggests only weak interaction with TM2 and TM3 segments. As already noted, the TM4 region has more hydrophobic residues (Leu, Ile, Val) that facilitate the highest degree of hydrophobicity compared to previously studied TM2 and TM3 segments [55]. According to our model, the BTL channel is organized as a pair of central α-helices (TM2 and TM3 segments), which regulates the uptake anions [17]. In that case, the other two TM fragments (TM4 together with the TM1) might play an essential role as external walls required for stabilizing the spatial organization of the whole channel.

## 4. Conclusions

The fourth transmembrane region of the BTL protein TM4 was predicted with the PredαTM algorithm using also with an improved statistical model. The stability of the predicted segments was assessed using molecular dynamics simulations performed in DPPC phospholipid. The results of the simulations indicated that the studied transmembrane region of BTL preserved a stable α-helical conformation.

Following our previous studies, we solved a high-resolution 3D structure of the fourth TM4 transmembrane fragment (Pro254–Ser276) in two widely used micellar media—SDS and DPC–using NMR spectroscopy. The structural analysis demonstrated the existence of an α-helical conformation in the central part of the TM4 segment in both surfactants. In the anionic SDS-$d_{25}$ micelle the α-helix includes the Pro258–Asn269 region. Use of the zwitterionic DPC-$d_{38}$ media changes the character of the α-helix to amphipathic which is marked by an additional turn and molecular dynamics processes in the low-frequency range. Besides, the snorkeling effect, that is, the interaction between positively charged groups in the lysine side chains and phosphate in the DPC micelle was confirmed with ${}^{31}$P relaxation measurements.

Our previous results obtained for TM2 and TM3 fragments demonstrated the possibility of tuning the carrier of various anions through the BTL channel using ‘open–close’ mechanism facilitated by *cis/trans* isomerization of the X–Pro peptide bonds contained the central prolines (Asn84–Pro85 in TM2 and Leu230–Pro231 in TM3) [17]. Taking into account more effective molecular dynamics in a low-frequency regime, the proposed relaxation mechanisms could be more pronounced in the case of zwitterionic micellar media.

The presented experimental data suggest that the last two transmembrane fragments (TM1 and TM4) probably form the outer wall of the BTL channel and act as an essential factor in the stabilization of the whole channel. How BTL facilitates bilirubin transport is currently not fully understood. Lack of structural BTL data along with the conformational flexibility observed for the bilirubin molecule hinders a more detailed insight into this process [56,57]. Since it is known that many human channels are 12-helix proteins the possibility that further oligomerisation of BTL is necessary to form an active channel cannot be ruled out. However, our initial computationally predicted arrangements of the BTL transmembrane regions TM1–TM4 [17] indicated that bilirubin could be in principle transported even via such an active channel. Thus, we believe that the acquired results will be helpful for the further exploration of the BTL functional mechanism, which could be of interest to drug discovery projects as well as to providing further atomistic insights leading to a better understanding of membrane proteins.

## 5. Materials and Methods

### 5.1. Prediction Algorithm of the TM4 Transmembrane Segment

The in-house algorithm PredαTM was used to predict the transmembrane regions of BTL from the primary sequence information [14,22]. The algorithm consists of a data-driven SVM classifier that was trained and validated to classify protein segments as either transmembrane or not. The protein segments in the dataset were mathematically characterized using 20-dimensional row sum vectors derived from their respective amino acid adjacency matrices. Accordingly, the transmembrane regions of BTL were predicted directly from the primary structure. The 340 residues long protein sequence was segmented into 329 partially overlapping segments, each 20 residues long. Then, the SVM classifier predicted each of the 329 segments to be either a transmembrane or non-transmembrane region.

The final structure of the transmembrane region was obtained from the initial prediction derived from the classifier refined using position-specific amino acid preference data. The statistical data was used to score all probable combinations of the terminal residues of BTL segments that form a transmembrane stretch as predicted by the classifier. One of the top three scoring segments that meet specific length and position criteria was reported as the final transmembrane region from that particular stretch.

To obtain a consensus, the following other state-of-art alpha helical transmembrane region predictors were also used to predict the transmembrane regions of BTL: TMpred [27], TopPred II [28], SOUSI [35], PRED-TMR [29], TMHMM [58], HMMTOP [32], Phobius [59], SVMtm [60], DAS-TMfilter [61], MEMSAT [62], SCAMPI [33], MemBrain [30], Philius [31], OCTOPUS [63] and TOPCONS [34]. This list includes a wide variety of algorithm types that use different sequence characterization methods and substantially cover the data space. The predictions from PredαTM and these 17 other algorithms were then compared.

#### Molecular Dynamics (MD) Simulation Procedure of the TM4 Fragment in DPPC Lipid

Molecular dynamics (MD) simulation studies were performed using the CHARMM molecular modeling suite [64]. CHARMM parameter and topology files (version 27) for proteins and lipids were utilized to specify the force field parameters of the protein and lipid DPPC molecules [65,66]. The system was first minimized in 1000 steps using the steepest descent (SD) method followed by 1000 steps of the modified Adopted Basis Newton-Raphson (ABNR) method. The equilibration stage used the scheme utilized in our previous studies resulting in a total of 375 ps simulation time. MD equilibration was performed in six steps utilizing a set of force constants provided in the Appendix A. The details of the MD equilibration procedure will be discussed further. In the first two equilibration steps, 1–2 the molecules were simulated for 25 ps by Langevin dynamics using one fs time step to avoid possible numerical integration issues. Next, four equilibration steps 3–6 were performed with standard MD procedure using a leapfrog integration algorithm.

In the next stage, a 2 fs step coupled with SHAKE algorithm was applied. The simulation times for steps 4–6 were 100 ps long. Production MD trajectories were also generated using leapfrog integration with a 2 fs simulation step and SHAKE. We carried out a 20 ns long MD simulation for both TM4 and TM4A. The results of the MD simulations were visualized and analyzed using VMD [36,67] and Gnuplot program [68].

### 5.2. Synthesis of the TM4 Peptide

Based on the results of MD simulations, synthetic peptide TM4 (Pro254–Ser276), corresponding to the TM4 transmembrane segments of BTL, were purchased from CASLO Laboratory, Denmark (www.caslo.com). The peptide was synthesized as trifluoroacetate salts and delivered as a lyophilized powder. Three lysine residues (a LysTag–KKK) were added at the N-termini to avoid the possible problems coming from the high hydrophobicity of the peptide. The amino acid sequence of synthetic TM4 is KKK${}^{254}$PNIFPLIACILLLSMNSTLSLFS${}^{276}$ with purity higher than 93.8%. Similar to our previous studies [17], we have introduced the ${}^{15}$N-labeled alanine into the central part of the TM4 peptide. Without the ${}^{15}$N-labeled Ala261, the study of the ${}^{15}$N NMR relaxation of the TM4 fragment would not be possible.

### 5.3. 3D Structure of the TM4 Fragment in SDS and DPC Micelles with NMR Spectroscopy

The NMR sample of TM4 segment in the SDS-$d_{25}$ (Sigma-Aldrich) was obtained by dissolving 1 mM peptide in 90% to 10% H${}_{2}$O/D${}_{2}$O ratio containing about 32 mg of SDS-$d_{25}$ (around 110 mM) with the addition of a small amount of DTT as a reducing agent. The concentration of SDS-$d_{25}$ was higher than critical micelle concentration (8.3 mM) at pH 6.5. Taking into account the mean aggregation number of SDS around 55, the SDS-$d_{25}$:TM4 molar ratio was kept at approximately 2:1, which means that statistically there are two micelles per one TM4 molecule [69].

The TM4 peptide in the DPC-$d_{38}$ was prepared according to the previously published protocol [70]. Namely, we have initially dissolved the TM4 peptide in 150 μL of deuterated methanol ($CD_{3}OD$, Sigma-Aldrich, Poznań, Poland). The obtained stock was intensely mixed for 20 min and diluted in 400 μL of the DPC-$d_{38}$:H${}_{2}$O/D${}_{2}$O solution. Similarly to the SDS-$d_{25}$ micelle the DPC-$d_{38}$:TM4 ratio was stabilized approximately 2:1. Before NMR experiments, each sample was placed in an ultrasound bath for 30 min to increase the homogeneity of the micelle:TM4 complex. As results the NMR samples of the TM4 peptide in SDS-$d_{25}$ and DPC-$d_{38}$ micelle demonstrated good homogenity and stability for a long time make them suitable to perform structural analysis with NMR spectroscopy (Appendix A).

The NMR datasets were acquired on Agilent DDR2 600 and Agilent DDR2 800 NMR spectrometers operated at 14.1 T (${}^{1}$H resonance frequency 599.98 MHz) and 18.8 T (${}^{1}$H resonance frequency 799.94 MHz). Both spectrometers equipped with three-channel DirectDrive console equipped with Performa IV *z*-gradient unit and HCN three-channel probe heads with an inverse detection. 2D homonuclear TOCSY (TOtal Correlation SpectroscopY) experiment was conducted with mixing time of 90 ms. A 2D NOESY (Nuclear Overhauser Effect SpectroscopY) [43] data set was acquired with 150 ms mixing time. The excitation sculpting DPFGSE pulse block was used to suppress the water signal in all acquired data [71]. The heteronuclear 2D ${}^{1}$H–${}^{15}$N and ${}^{1}$H–${}^{13}$C HSQC spectra were recorded on 18.8 T (Agilent DDR2 800) on natural abundance of the ${}^{13}$C and ${}^{15}$N isotopes. All chemical shifts were referenced with respect to external sodium 2,2-dimethyl-2-silapentane-5-sulfonate (DSS) using Ξ = 0.251449530 and 0.101329118 ratio for indirectly referenced ${}^{13}$C and ${}^{15}$N resonances, respectively [72]. The recorded NMR data were processed using NMRPipe software [73] and analyzed with the Sparky program [74].

The TM4 fragment contain two prolines near the N-termini, in positions 254 and 258. The conformation for X–Pro peptide bonds (Lys–Pro254 and Phe257–Pro258) were deduced from chemical shifts for the ${}^{13}$C${}^{\beta }$ and ${}^{13}$C${}^{\gamma }$ resonances and defined as *trans* in both surfactants [75]. This was later confirmed by observation of ${}^{1}$H${}^{\alpha }$–${}^{1}$H${}^{\delta }$ cross-peaks on ${}^{1}$H–${}^{1}$H NOESY spectra. The reduced state of the thiol group in Cys262 were confirmed with the chemical shifts (27.6 and 29.3 ppm) assigned to the ${}^{13}$C${}^{\beta }$ resonances [76] in the SDS-$d_{25}$ and DPC-$d_{38}$ micelle, respectively.

Molecular dynamics simulations of the TM4 peptide in SDS and DPC micelle were conducted with the AMBER 11 or AMBER 14 software packages [77,78] using the *param99* force field. For both types of surfactants, the molecular dynamics simulations were performed similarly. The starting coordinates of the SDS–water or DPC–water system were taken from previous simulations and, if necessary, the topology was adapted to AMBER. The simulations were started by positioning the TM4 fragment into the simulation box with its center of mass coinciding with that of the SDS or DPC micelle. The chloride ions were added to neutralize a total charge +3 of the system. To remove the bad initial contact between peptide and micelle core and to prevent penetration of water during equilibration the peptide and bulk water were kept under weak harmonic constraints with force constants of 10 and 5 kcal/(mol × Å), respectively. Those constraints were removed after 20,000 steps of minimization (steepest descent method). Later, the entire system was minimized for 20,000 steps without any constraints. After that, the TM4: micelle complex was subjected to dynamics under constant pressure with an average pressure of 1 atm and temperature (303 K) for 36–40 ns with time-averaged distance restraints (TAV) derived from NMR data. The interproton distances were introduced with the force constants f = 50 kcal/(mol × Å${}^{2}$). The initial conformation of the TM4 peptide was taken as the output of the CYANA calculations. The orientation of the peptide is not important due to the spherical symmetry of micelles. Taking into account the pH of the NMR sample, the side chains of three lysines on N-terminus were defined as protonated and have to be neutralized with chloride ions.

The obtained 3D structures were analyzed with Ptraj program included in the AMBER package [78]. The interactions of the peptide with the micelle were characterized by calculation of the radial distribution functions (RDF) for TM4 peptide side chains and polar headgroups, the hydrophobic part of the micelle and water. The RDF data were evaluated on average over the last 200 ps of MD simulations. The final analysis and visualization were performed with either MOLMOL [79] or Chimera [80] software.

### 5.4. ${}^{1}$H, ${}^{2}$H and ${}^{31}$P PGSE Diffusion Measurements of the TM4 Fragment in SDS-$d_{25}$ and DPC-$d_{38}$ Surfactants

The experimental data on the binding of the TM4 fragment to the SDS-$d_{25}$ or DPC-$d_{38}$ micelles were obtained utilizing Pulsed Field Gradient Spin Echo NMR (PGSE NMR). The experiments were carried out in the temperature range 288–313 K. The experimental data on the ${}^{1}$H were acquired on an Agilent VNMRS 800 NMR spectrometer with a Performa IV gradient unit forming up to 60 Gs/cm gradient pulses along *z*-direction. The $D_{tr}$ were accumulated as 512 accumulations with 25 gradient steps for TM4 in both SDS-$d_{25}$ or DPC-$d_{38}$ surfactants. The diffusion coefficients of micelles were obtained from ${}^{2}$H and ${}^{31}$P NMR experiments using an Agilent DDR2 600 MHz spectrometer equipped with DOTY DSI-1372 X-H probehead capable of high magnetic field gradients (up to 2800 Gs/m). The DPFGDSTE pulse sequence was used (Double Pulsed Field Gradient Double Stimulated Echo) [81,82] and the gradient amplitudes were varied between 9 and 826 (Gs/cm). Keeping in mind, that SDS and DPC micelles were perdeuterated, self-diffusion coefficients $D_{tr}$ of the SDS-$d_{25}$ and DPC-$d_{38}$ micelles were extracted from data acquired on the ${}^{2}$H (Figure 4) and ${}^{31}$P (DPC-$d_{38}$ only).

The processing of diffusion data was performed using the CONTIN algorithm [83] embedded in VnmrJ 4.2 (Agilent Inc.,Santa Clara, CA, USA) software to trace eventual polydispersity of diffusion coefficients. The $D_{tr}$ values were obtained from the gradient attenuated integral values of the resonances according to the Stejskal-Tanner equation [84]:$$I=I_{0}\exp\left(-D{\left(G\gamma \delta \right)}^{2}\left(\Delta -\delta /3\right)\right)$$
where γ is the ${}^{1}$H, ${}^{2}$H or ${}^{31}$P gyromagnetic ratio, δ is time of gradient duration, Δ is diffusion time and *G* is the gradient strength [84]. The experimental data points were recorded for the following diffusion times Δ: 40 ms for ${}^{1}$H, 60 ms for ${}^{31}$P and 200 ms for ${}^{2}$H respectively.

### 5.5. Acquisition of ${}^{15}$N and ${}^{31}$P NMR Relaxation Data

${}^{15}$N relaxation measurements were performed at a 303 K on the magnetic field strength of 14.1 T. The ${}^{15}$N longitudinal ($R_{1}$) and transverse ($R_{2}$) relaxation rates for ${}^{15}$N-labeled Ala261 were recorded according to the previously published pulse sequences [85] included in BioPack (Agilent Inc., USA) software. The ${}^{15}$N $R_{1}$ data sets were acquired with ten evolution times: 10, 90, 170, 290, 410, 550, 690, 850, 1010 and 1250 ms. The ${}^{15}$N $R_{2}$ experiments were performed using Carr-Purcell-Meiboom-Gill (CPMG) pulse train using the refocusing time of 650 μs with eight experimental points: 10, 30 50, 70, 90, 130, 170 and 210 ms. To suppress the effect of the cross-correlation, the delays between π (${}^{1}$H) pulses were 5 and 10 ms for acquiring the $R_{1}$ and $R_{2}$ data, respectively [86].

The recycle delay in all relaxation experiments was kept as long as 3.5 s. The ${}^{15}$N relaxation rates were extracted from the nonlinear two-parameter fit of peak amplitudes as a single exponent decay using the program Gnuplot 4.0 (www.gnuplot.info). Errors in the $R_{1}$ and $R_{2}$ values were determined from the variance–covariation matrix.

The ${}^{31}$P longitudinal ($R_{1}$) and transversal ($R_{2}$) relaxation rates were measured for the ${}^{31}$P isotope in the phosphate group PO_4_ with and without the TM4 fragment in the DPC-$d_{38}$ micelle. The $R_{1}$ value was extracted from the inversion recovery experiment acquired with 10 points (0.0625, 0.125, 0.25, 0.5, 1, 2, 4, 8, 16 and 32 s). The $R_{2}$ relaxation rate was evaluated with π/2${}_{x}$–$\pi_{y}$ pulse sequence on base 9 points (1.25, 2.5, 5, 10, 20, 40, 80, 160 and 320 ms). The recycling delay was 20 s in $R_{1}$ and 5 s in $R_{2}$ measurement.

To explore the slow dynamic processes in the DPC micelle, the ${}^{31}$P $R_{2}$ CPMG experiment π/2${}_{x}$–[$\tau_{cp}$/2–$\pi_{y}$–$\tau_{cp}$–echo]${}_{n}$ was performed twice. For the empty DPC-$d_{38}$ micelle data were acquired with 10 $\tau_{CP}$ delays–50 μs, 80 μs, 0.1 ms, 0.3 ms, 0.5 ms, 0.9 ms, 1.0 ms, 1.4 ms, 1.5 ms and 2.0 ms. Where the TM4 fragment was in DPC-$d_{38}$ the CPMG data contained 17 delays: 50 μs, 60 μs, 70 μs, 80 μs, 0.1 ms, 0.2 ms (twice), 0.5 ms, 0.9 ms (twice), 1.1 ms (twice), 1.2 ms, 1.3 ms, 1.4 ms, 1.5 ms and 2.0 ms. In both cases the amplitudes of spin-echoes were analyzed with the Carver-Richards (CR) model [52] described by a two–site conformational exchange processes (See Appendix A for details) [51].

## Figures and Tables

**Figure 1 ijms-20-04172-f001:**
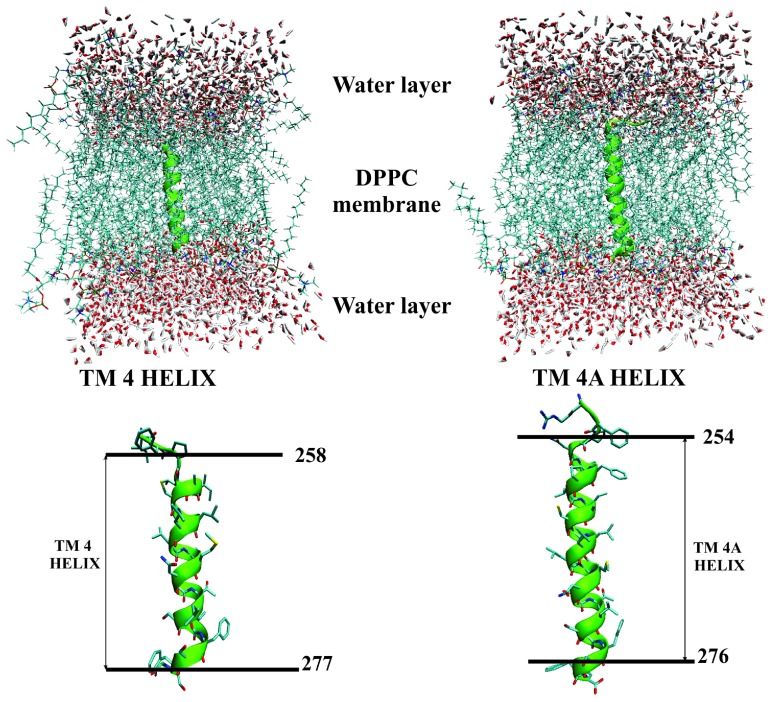
Simulation models of TM4 (**left**) and TM4A (**right**) transmembrane bilitranslocase (BTL) segments inserted into the dipalmitoylphosphatidylcholine (DPPC) membrane in the aqueous medium. Representative snapshots of the TM4 and TM4A α-helices with approximate borders of the helical structures depicted.

**Figure 2 ijms-20-04172-f002:**
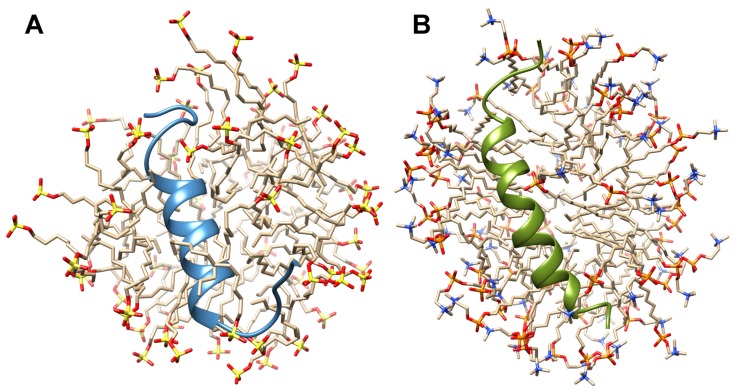
High-resolution 3D structures of TM4 segment (Pro252–Ser276) evaluated on the basis of acquired nuclear magnetic resonance (NMR) data and molecular dynamics simulations in SDS-$d_{25}$ (**A**) and DPC-$d_{38}$ (**B**) micelle.

**Figure 3 ijms-20-04172-f003:**
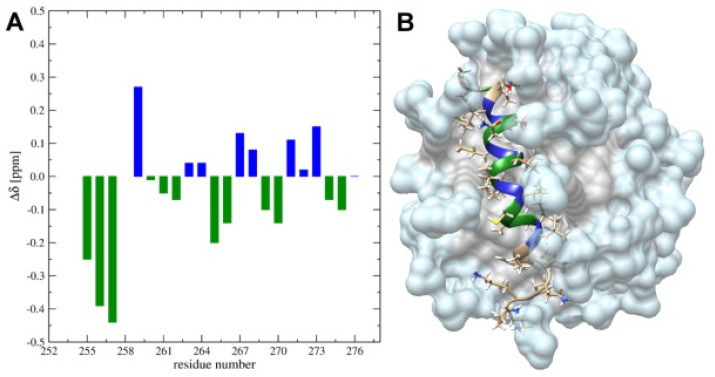
(**A**) CSP (chemical shift perturbation) plot for the amide (${}^{1}$H${}^{N}$) protons obtained for SDS-$d_{25}$ and DPC-$d_{38}$ media. (**B**) The 3D structure of the TM4 fragment in the DPC-$d_{38}$ micelle. Residues demonstrated upfield and downfield shifts of the ${}^{1}$H${}^{N}$ resonances in DPC-$d_{38}$ micelle compare to SDS-$d_{25}$ are highlighted as blue and green, respectively.

**Figure 4 ijms-20-04172-f004:**
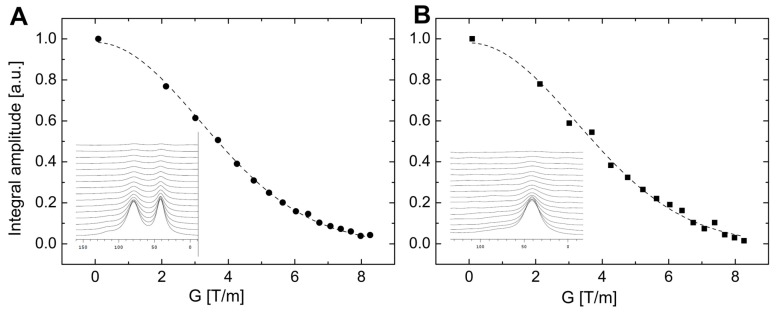
The ${}^{2}$H PGSE spin echo attenuation obtained at 303 K for SDS-$d_{25}$ (**A**) and DPC-$d_{38}$ (**B**) micelles contained the TM4 peptide. The ${}^{2}$H 1D NMR spectra used for extract $D_{tr}$ values are presented as insets.

**Figure 5 ijms-20-04172-f005:**
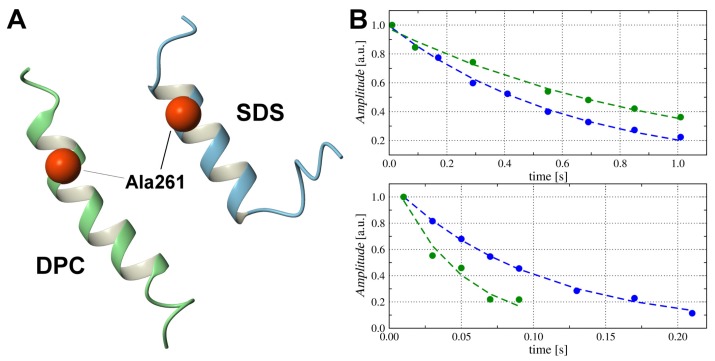
(**A**) The 3D structure of TM4 fragment in SDS-$d_{25}$ and DPC-$d_{38}$ surfactants is presented. The position of ${}^{15}$N nuclei in ${}^{15}$N-labeled Ala261 are depicted as balls. (**B**) Experimental $R_{1}$ (up) and $R_{2}$ (down) relaxation decay recorded at 18.1 T for ${}^{15}$N-labeled Ala261 inside SDS-$d_{25}$ (blue) and DPC-$d_{38}$ (green) micelle.

**Figure 6 ijms-20-04172-f006:**
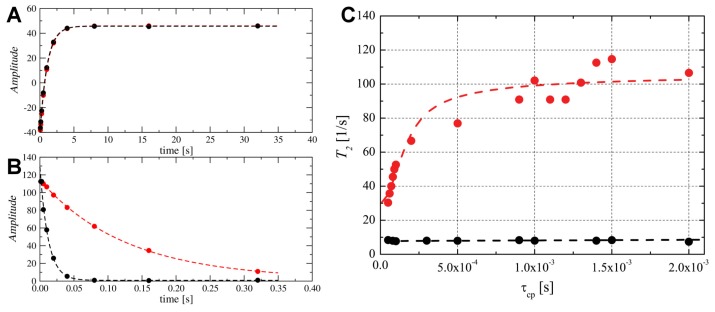
The ${}^{31}$P relaxation experiments acquired for the phosphate group in the DPC micelle without (black) and with (red) TM4 segment. (**A**) ${}^{31}$P longitudinal $R_{1}$ relaxation recovery and (**B**) transversal $R_{2}$ relaxation decay curves. (**C**) ${}^{31}$P CPMG dispersion NMR experiment. The best fit obtained with CR model for DPC-$d_{38}$ micelle with TM4 segment is presented.

**Figure 7 ijms-20-04172-f007:**
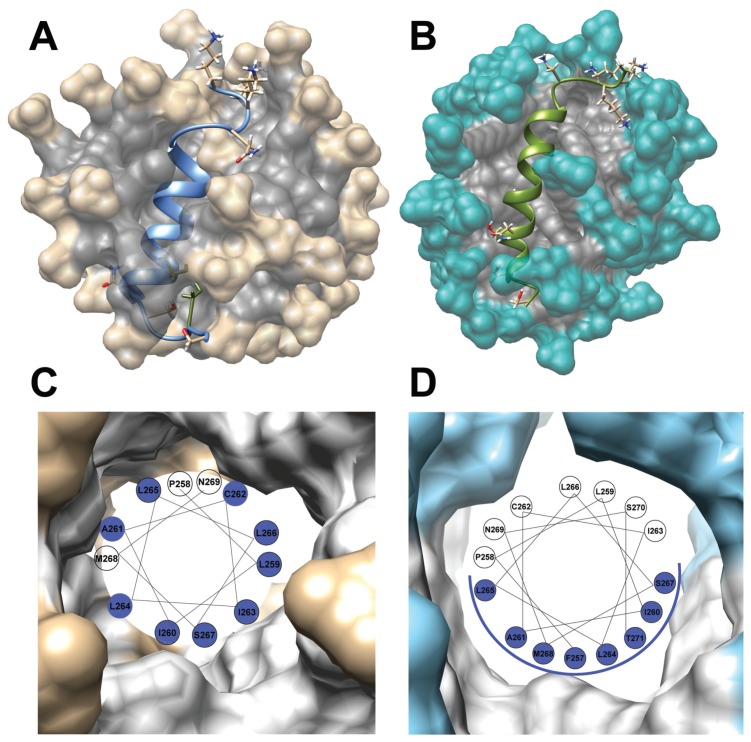
The 3D structure of TM4 fragment in SDS-$d_{25}$ (**A**) and DPC-$d_{38}$ (**B**) presented with hydrophobic (dark) and hydrophilic core of micelle. Visualization of the radial distribution function (RDF) data by a wheel plot of the TM4 α-helix inside SDS-$d_{25}$ (**C**) and DPC-$d_{38}$ (**D**) micelle, respectively. The residues with side chains located in the hydrophobic environment are highlighted in blue.

**Table 1 ijms-20-04172-t001:** ${}^{15}$N $R_{1}$ and $R_{2}$ relaxation rates extracted from experimental data measured for the ${}^{15}$N-labeled Ala261 at 303K on 18.8 T magnetic field.

Media	*R*${}_{1}$ (s${}^{-1}$)	*R*${}_{2}$ (s${}^{-1}$)	$\tau_{c}$ (ns) ${}^{a}$
SDS-*d*${}_{25}$	1.59 ± 0.08	10.01 ± 0.81	7.26 ± 0.41
DPC-*d*${}_{38}$	0.68 ± 0.01	28.58 ± 1.09	20.56 ± 0.41

${}^{a}$ Overall correlation times ($\tau_{c}$) calculated from the ${}^{15}$N *R*${}_{2}$/*R*${}_{1}$ ratio assuming isotropic rotation model of the TM4 segment.

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
