# Peer review of "Structural Analysis and Dynamic Processes of the Transmembrane Segment Inside Different Micellar Environments—Implications for the TM4 Fragment of the Bilitranslocase Protein"

_ijms, 2019, doi:10.3390/ijms20174172_

Round 1

Reviewer 1 Report

In this paper, the authors have elucidated the structure of bilitranslocase (BTL) TM4 region using two different surfactants (anionic SDS and zwitterionic DPC) via NMR spectroscopy and MD simulations in DPPC lipid. Results showed that TM4 adopted a stable α-helical conformation in the central part of the TM4 region in both surfactants. Previously, the authors have determined the structures of TM2 and TM3 of BTL using a similar protocol which adds value to this study. Overall, the manuscript is well presented and the significance of BTL TM4 analysis is clear. Besides the minor issues listed below, this paper could be accepted for publication in IJMS.

Minor comments:

Overall, this manuscript needs intensive English editing. Several non-idiomatic sentences and grammatical errors are found throughout the manuscript. Please edit the entire manuscript with a native speaker. Line 20, change keyword ‘trans-membrane’ to ‘transmembrane.’ Check the usage of commas and hyphens in the entire manuscript. Define TM4A in its first appearance in the Introduction section (lines 56-57). Lines 60-61, statement seems ambiguous. Rephrase it. Similarly, rephrase the sentences in lines 66-70. Lines 114-119, since the term TM2, TM3 or TM4 is well understood, there is no need to mention as second transmembrane, third transmembrane and so on. I think that it would be better if authors could provide a separate graphical figure showing TM2, TM3, and TM4 segments together for clarity to readers and insert it near the discussion section, since they have previously elucidated the structures of TM3 and TM4. Lines 384-386, rephrase the sentence. Check the spelling of ‘computationally’ in line 388.

Author Response

Overall remark

Overall, this manuscript needs intensive English editing. Several non-idiomatic sentences and grammatical errors are found throughout the manuscript. Please edit the entire manuscript with a native speaker.

Response:

We perform intensive language editing of the whole manuscript by a qualified editor. Numerous corrections and improvements were provided to the revised version.

Point 1

Line 20, change keyword ‘trans-membrane’ to ‘transmembrane.’

Response 1:

We correct keyword ‘trans-membrane’ to ‘transmembrane’.

Point 2

Define TM4A in its first appearance in the Introduction section (lines 56-57).

Response 2:

The following sentence was modified according to your remarks (lines 57-59 revised manuscript)

from: “Performing structural analysis of the TM4 / TM4A segment in aqueous solutions of DPPC ...”

to: “Performing structural analysis of the two possible lengths of the TM4 segment, defined as TM4, predicted by our initial algorithm, and TM4A obtained by its statistically improved version, in aqueous solution of DPPC ...”

Point 3

Lines 60-61, statement seems ambiguous. Rephrase it.

Response 3:

The following sentence was modified according to your remarks (lines 62-63 revised manuscript)

from: “The micelles are formed by the components jointed two distinct parts with different affinity to a solvent – hydrophilic ’head’ and hydrophobic ’tail’.”

to: “The micelar environments are formed by the components with different affinity to a solvent – hydrophilic ’head’ and hydrophobic ’tail’.”

Point 4

Similarly, rephrase the sentences in lines 66-70.

Response 4:

The following sentence was modified according to your remarks (lines 66-71 revised manuscript)

from: “It is constitute better model for the eukaryotic cell membrane [24] compared to SDS, altrough helical membrane proteins (like BTL) demonstrate lower stability in zwitterionic lipid media due to the highly hydrophobic composition of transmembrane fragments.[25]”

to: “The zwitterionic dodecylphosphocholine (DPC), belonged to the class of alkyl phosphocholine detergents constitutes a better model for the eukaryotic cell membrane [24] compared to SDS.

Helical membrane proteins (like BTL) demonstrate lower stability in zwitterionic lipid media due to the highly hydrophobic composition of transmembrane fragments.[25]”

Point 5

Lines 114-119, since the term TM2, TM3 or TM4 is well understood, there is no need to mention as second transmembrane, third transmembrane and so on.

Response 5:

We remove additional definition on TM fragments as second, third, etc in the fragment which you mention. The following sentence was modified according to your remarks (lines 117-122 revised manuscript)

from: “In conclusion, while most predictors have failed to identify the second and the third transmembrane regions TM2 and TM3, the transmembrane regions TM1 and TM4 were predicted most consistently. Interestingly, the content of leucine in the TM1 and TM4 segments was 28% and 26% respectively. In both cases, the predicted content of Leu is significantly higher than the average for α transmembrane regions. On the other hand, the second TM2 and the third TM3 transmembrane regions show a significantly lower occurrence of Leu, 15%, and 10.5%, respectively.”

to: “In conclusion, while most predictors have failed to identify the TM2 and TM3 transmembrane regions, the TM1 and TM4 were predicted most consistently. Interestingly, the content of leucine in the TM1 and TM4 segments was 28% and 26% respectively. In both cases, the predicted content of Leu is significantly higher than the average for α transmembrane regions. On the other hand, the TM2 and the TM3 show a significantly lower occurrence of Leu, 15%, and 10.5%, respectively”

Point 6

I think that it would be better if authors could provide a separate graphical figure showing TM2, TM3, and TM4 segments together for clarity to readers and insert it near the discussion section, since they have previously elucidated the structures of TM3 and TM4.

Response 6:

We provided the graphical representation of 3D structures of the TM2, TM3 and TM4 fragments solved in SDS micelle as Figure S15 in Supporting Materials. It referenced in Discussion section “A comparison with previously evaluated structures of the TM2 and TM3 transmembrane fragments in SDS micelle” (lines 341, 343 and 349 revised manuscript).

Point 7

Lines 384-386, rephrase the sentence. Check the spelling of ‘computationally’ in line 388.

Response 7:

The following sentence was modified according to your remarks (lines 387-391 revised manuscript)

from: “The process of bilirubin transport via BTL is currently very limited understood, especially due to the lack of structural data and the conformational flexibility of the bilirubin molecule.[56,57]”

to: How BTL facilitates the bilirubin transport is currently not fully understood. Lack of structural BTL data along with the conformational flexibility observed for the bilirubin molecule hinder a more detailed insight into this process.[56,57]”

The typo ‘computatinally’ was corrected and the word now spells ‘computationally’.

Reviewer 2 Report

The manuscript presented by Szutkowski and co-workers describes structural-dynamic investigations of predicted TM4 domain of the BTL channel. In contrast to their previous works with other BTL TM segments embedded into SDS micelles, the current results were obtained using NMR spectroscopy and Molecular Dynamics in two different micellar media, anionic SDS and zwitterionic DPC. Such approach allowed to obtain a detailed comparison of numerous parameters in different membrane-mimicking environments that makes the work is methodologically notable for the structural biology of membrane proteins. Nevertheless, the manuscript itself does not a mature publication, and I have the following comments, which need to be addressed.

1) Does the NMR-derived hydrodynamic radius of micelles containing TM4 correspond to the DPC and SDS micelle sizes simulated by MD?

2) The overall correlation time estimated by NMR for TM4 in DPC is unexpectedly high and exceeds 20 ns. Did the authors consider the oligomerization of TM4 in DPC micellar environment?

3) Spin diffusion process must be observed in the NOESY spectra acquired with 150-ms mixing time at such high overall correlation times that could distort the calculated structures of TM4 in both micellar media.

4) The NOESY mixing times indicated in the captions of figures S14 and S15 are different from ones in the “Materials and Methods” section.

5) Similar to Figure S1, please show the behavior of the secondary structure along the MD trace of TM4 in SDS and DPS micelles.

6) There are some mistakes in the manuscript text (e.g. line 219). Figures 3 and S1 have low resolution. Axes on some graphs are poorly marked or even not signed (e.g. Figure S9).

Author Response

Point 1

Does the NMR-derived hydrodynamic radius of micelles containing TM4 correspond to the DPC and SDS micelle sizes simulated by MD?

Response 1:

Generally speaking no. In our study, we solved the 3D structure of the TM segment. In MD simulations we used SDS and DPC micelle previously parametrized in the AMBER 14 force field contained 64 monomers. According to our estimation, the number of aggregation is higher than used in the MD trajectory. So, in our MD simulations we can speak about the “micellar environment”, but a direct comparison with an experimentally determined hydrodynamic radius of micelle rather not possible.

Point 2

The overall correlation time estimated by NMR for TM4 in DPC is unexpectedly high and exceeds 20 ns. Did the authors consider the oligomerization of TM4 in DPC micellar environment?

Response 2:

Yes, it was our first idea. This was the reason why we put so many efforts to perform diffusion experiments of different nuclei. As a result, the translational diffusion coefficients do not confirm the effect of the oligomerization of TM4 fragments in DPC micelle. At the moment, we couldn’t explain so big difference between rotational and translational motions of TM4 peptide in DPC. Probably, we have an effect of the amphipathic character of the central helix additionally amplified by snorkeling. It’s interesting to see if such effects will be observed for TM2 and TM3 fragments, which were previously studied in SDS micelle in our group. This work is currently in progress.

Point 3

Spin diffusion process must be observed in the NOESY spectra acquired with 150-ms mixing time at such high overall correlation times that could distort the calculated structures of TM4 in both micellar media.

Response 3:

We’re recorded a couple of NOESY spectra for TM4 fragment in both surfactants with mixing from 90 ms up to 200 ms. The idea was not to evaluate the build-up curve but solve problems with resonance assignments in side-chains, which were not visible. According to our experience, the NOESY experiment with long mixing time (strong spin-diffusion effect) sometimes demonstrates more signals compare to the TOCSY. The assignment procedure of TM4 peptide contained 6 leucines and 3 isoleucines were not trivial in SDS and DPC. The 3D structures evaluation were performed on distance constraints yielded from jointed analysis NOESY spectra collected at 150 and 120 ms. At the first stage, the initial 3D structures were calculated with cyana software and we detect no violations greater than 0.2 A. After that, that structures were refined with AMBER where we did not observe structural distortions.

Point 4

The NOESY mixing times indicated in the captions of figures S14 and S15 are different from ones in the “Materials and Methods” section.

Response 4:

As we note in answer to the previous remark, the NOESY spectra with various mixing time were collected and analyzed. Figures S14 and S15 demonstrated the quality of experimental data used for 3D structure evaluation.

Point 5

Similar to Figure S1, please show the behavior of the secondary structure along the MD trace of TM4 in SDS and DPS micelles.

Response 5:

We provided the time plot of secondary structure analysis for the TM4 segment in SDS and DPC micelle is presented in Supporting Materials (Figure S12).

Point 6

There are some mistakes in the manuscript text (e.g. line 219). Figures 3 and S1 have low resolution. Axes on some graphs are poorly marked or even not signed (e.g. Figure S9).

Response 6:

We provide the correct reference to Figure 3B on line 219. The new version of Figures 3 and Figure S1 with higher resolution are included in the revised version of the manuscript. The axis in Figure S9 is provided. The marked axis in other Figures in the manuscript is improved.